# An Adjacency Encoding Information-Based Fast Affine Motion Estimation Method for Versatile Video Coding

**Ximei Li, Jun He \*, Qi Li and Xingru Chen**

Huaihua Polytechnic College, Huaihua 418000, China
**\*** Correspondence: hjun20220808@163.com

**Abstract:** Versatile video coding (VVC), a new generation video coding standard, achieves significant improvements over high efficiency video coding (HEVC) due to its added advanced coding tools. Despite the fact that affine motion estimation adopted in VVC takes into account the translational, rotational, and scaling motions of the object to improve the accuracy of interprediction, this technique adds a high computational complexity, making VVC unsuitable for use in real-time applications. To address this issue, an adjacency encoding information-based fast affine motion estimation method for VVC is proposed in this paper. First, this paper counts the probability of using the affine mode in interprediction. Then we analyze the trade-off between computational complexity and performance improvement based on statistical information. Finally, by exploring the mutual exclusivity between skip and affine modes, an enhanced method is proposed to reduce interprediction complexity. Experimental results show that compared with the VVC, the proposed low-complexity method achieves 10.11% total encoding time reduction and 40.85% time saving of affine motion estimation with a 0.16% Bjøontegaard delta bitrate (BDBR) increase.

**Keywords:** versatile video coding; inter-prediction; affine motion estimation; low-complexity





## 1. Introduction

The rapid development of new video applications, together with emerging videos in high frame rate (HFR), high dynamic range (HDR), and high resolution, raises an urgent demand to develop a new generation of video coding standards with coding efficiency beyond the high efficiency video coding (HEVC) standard [1]. The latest standard, versatile video coding [2–4], is launched by the joint video experts team (JVET) to solve the above issue. The VVC confirmed in July 2020 achieves superior encoding performance to HEVC by adopting a series of coding tools with high computation [5–8]. For interprediction, the decoder-side motion vector refinement (DMVR) [9], bi-directional optical flow (BDOF) [10, 11], and affine motion compensation (AMC) [12–14] are used to optimize the accuracy of prediction. Furthermore, the cross-component linear model (CCLM) [15,16] and the position-dependent intraprediction combination (PDPC) [17,18] are adopted to enhance the intraprediction. In order to further eliminate frequency redundancy, the low-frequency non-separable transform (LFNST) [19,20] is employed in VVC.

As an important coding tool in VVC, the affine motion model improves interprediction accuracy. Compared with the previous motion estimation models, the affine motion model takes into account the translational motion and the rotation and scaling of the object, which is more consistent with the trajectory of the object in real life. There are two types of affine motion models in VVC, the four-parameter affine motion model and the six-parameter affine motion model. Under different models, the affine motion vectors of the current coding unit (CU) are calculated from the corresponding control points. Figure 1 shows the affine motion models in VVC. Specifically, Figure 1a,b are the four-parameter affine motion model and the six-parameter affine motion model, respectively.

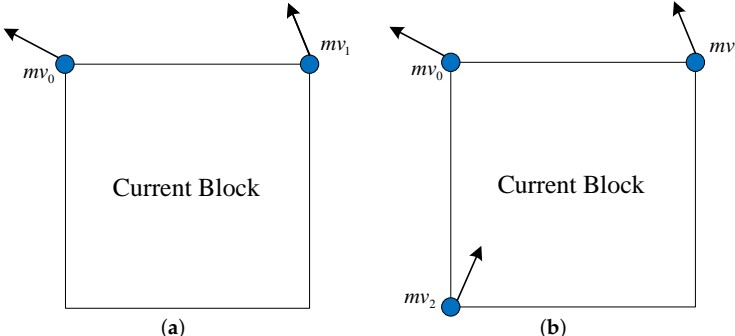

**Figure 1.** The affine motion models in VVC. (**a**) Four-parameter affine motion model; (**b**) six-parameter affine motion model.

Here, we successively describe the specific motion estimation process of the four-parameter and six-parameter affine motion models. In Figure 1a, based on the control point motion vectors (CPMV) at the top-left location $mv_0$ and top-right location $mv_1$, the motion vector (MV) of the current block centered on pixel $(x, y)$ can be calculated by

$$
\begin{cases}
mv^h(x,y) = \dfrac{mv_1^h - mv_0^h}{W}x + \dfrac{mv_1^v - mv_0^v}{W}y + mv_0^h, \\
mv^v(x,y) = \dfrac{mv_1^v - mv_0^v}{W}x + \dfrac{mv_1^h - mv_0^h}{W}y + mv_0^v,
\end{cases}
\tag{1}
$$

where $mv^h(x,y)$ and $mv^v(x,y)$ represent the horizontal and vertical motion vectors of the current block. $mv_0^h$, and $mv_0^v$ are the horizontal and vertical vectors of CPMV at the top-left location. $mv_1^h$ and $mv_1^v$ denote the horizontal and vertical vectors of CPMV in the top-right location. $W$ is the width of the current block.

For the six-parameter affine motion model in Figure 1b, it has one more CPMV at the bottom-left location $mv_2$ than the four-parameter affine motion model. The calculation of the MV of the current block is shown as follows,

$$
\begin{cases}
mv^h(x,y) = \dfrac{mv_1^h - mv_0^h}{W}x + \dfrac{mv_2^v - mv_0^v}{H}y + mv_0^h, \\
mv^v(x,y) = \dfrac{mv_1^v - mv_0^v}{W}x + \dfrac{mv_2^h - mv_0^h}{H}y + mv_0^v,
\end{cases}
\tag{2}
$$

where $mv_2^h$ and $mv_2^v$ are the horizontal and vertical vectors of CPMV at the bottom-left location. $H$ represents the height of the current block.

Although the introduction of the affine motion model improves the encoding performance of the VVC, it brings a huge computational complexity to the encoder of VVC. Furthermore, a variety of other coding tools are also adopted in VVC to obtain a better coding performance, such as multi-type tree partition and the 67 intraprediction modes. These coding tools with complex computation make the encoding process in VVC more flexible but also more complex than HEVC, which makes VVC difficult to use for real-time applications. Hence, it is necessary to simplify the encoding process in VVC to make it suitable for hardware devices.

In this paper, we proposed an adjacency encoding information-based fast affine motion estimation method to accelerate the interprediction process. The proposed method can also be combined with the fast CU partition method to save more encoding time.

The primary contributions of this work are summarized as follows:

1.  Distinguishing from most of the previous fast algorithms that focus on the CU partition, we fully explore the affine motion estimation in interprediction and propose a fast affine motion estimation algorithm based on the adjacency encoding information to achieve the savings of encoding time.

2.  We count the proportion of CUs that use affine mode as the best interprediction in test sequences with different resolutions. Then we analyze the trade-off between computational complexity and performance improvement based on statistical information.
3.  The affine motion estimation skipping method is proposed by exploring the relationship between affine and skip modes in interprediction.

The rest of this paper is arranged as follows: Section 2 reviews some related studies by focusing on the fast algorithm for VVC. Section 3 introduces the proposed adjacency encoding information-based fast affine motion estimation method in detail. The experimental results and analysis are presented in Section 4. Finally, the conclusion of this paper is given in Section 5.

## 2. Related Work

Typically, an encoder with low computational complexity speeds up the encoding and transmission of video, resulting in low latency video streams. Therefore, designing video encoders with high coding efficiency and low complexity is a core requirement for real-time applications with limited transmission bandwidth and computing power.

Although some studies have been done to reduce the computational complexity of VVC, most of these works have focused on making early decisions to speed up the partitioning process [21–27]. Min et al. [22] proposed a fast method to determine CU partitioning by exploiting global and local edge complexities in multiple directions. Zhao et al. [23] extracted the standard deviation and the edge point ratio to speed up the CU partition. The CU splitting information and the temporal location of the coded frame were used in [24] for the low-complexity encoder. In [25], the edge information extracted by the canny operator was utilized for CU partition in the intra- and intercoding. Similarly, the spatial features were used in [26] to reduce the computational complexity of the CU binary tree partition process. Lei et al. [27] introduced a fast method to accelerate the encoding process by exploring the content property. In recent years, some fast methods [28–31] based on a convolutional neural network (CNN) to extract and utilize features have been proposed. In [32], Wu et al. proposed a hierarchy grid fully convolutional network (HG-FCN) framework for fast VVC intra coding. In addition, there are some studies to speed up the encoding process by reducing the computational complexity of other coding tools. Pan et al. [33] introduced an entropy-based algorithm for rate-distortion optimization (RDO) mode decision. The histogram of oriented gradient features and the depth information were jointed in [34] to reduce the computational complexity in the visual sensor networks (VSNs). However, a few studies focus on fast algorithms for interprediction. In [35], a low-complexity method combining multi-type tree partition was proposed to reduce the computation of multiple transform selection. In [36], Ren et al. proposed an advanced fast interprediction method based on edge detection. Jung et al. [37] proposed a context-based inter mode decision method to accelerate the encoding process. There is still much room for improvement in reducing the coding complexity of VVC.

This paper focuses on reducing the computational complexity of the interprediction in VVC to speed up the encoding process and meet the requirements of real-time applications. It is worth mentioning that the proposed fast affine motion estimation method can also be combined with fast CU partition methods to save the encoding time further.

## 3. Materials and Methods

To speed up the interprediction in VVC encoding process, an adjacency encoding information-based fast affine motion estimation method is introduced in this paper. First, we trade off the computational complexity and performance improvement of affine motion estimation. Secondly, the adjacency encoding information is used to determine whether to skip the affine mode in the interprediction. The details are described as follows.

### 3.1. Statistics and Analysis of the Adjacency Encoding Information

In general, the moving objects in the video tend to occupy a small part of the whole frame, whereas the rest is mostly the background area. The trajectory of moving objects in a video is panning, rotating, and zooming, and the background area is in translational movement or stationary. In the interprediction process of VVC, the affine mode is performed for all blocks in the frame. Only a small proportion of the blocks choose the affine mode as the best interprediction. As a result, the process of affine motion estimation for most blocks is redundant, which significantly increases the computational complexity and encoding time. To trade off added computational complexity and encoding performance gains, we have counted the percentage of blocks that used the affine mode as the best interprediction in test videos of different resolutions to obtain a more accurate usage rate. Specifically, we assume that event $A$ represents the current block choosing the affine mode as the optimal interprediction. $P(A)$ denotes the probability that blocks in which the affine mode as the optimal interprediction is selected and can be calculated as

$$P(A) = \frac{C_{affine}}{C_{total}} \times 100\%, \tag{3}$$

where $C_{affine}$ represents the blocks selecting the affine mode as the best interprediction. $C_{total}$ is the total number of the blocks. The statistical results of the proportion of blocks selecting affine intermode are shown in Table 1. We can observe that the proportion of blocks that select affine mode is relatively small. The average value is 12.3%. This illustrates that only a small part of the region finally selects an affine mode in the interprediction of blocks. However, when choosing the optimal interprediction, the affine motion estimation is calculated for each block, which significantly increases the computational complexity. The additional unnecessary calculations create a large waste in terms of encoding time. Combining the above analysis, we propose a method which reduces the encoding time without significantly degrading the encoding performance to trade-off computational complexity and encoding performance.

**Table 1.** Statistical information of the use of affine mode as optimal interprediction.

| Sequences | $P(A)$ |
|---|---|
| BlowingBubbles $416 \times 240$ | 10.7% |
| BQMall $832 \times 480$ | 10.3% |
| RaceHorsesC $832 \times 480$ | 9.8% |
| FourPeople $1280 \times 720$ | 9.0% |
| Cactus $1920 \times 1080$ | 12.2% |
| Campfire $3840 \times 2160$ | 15.9% |
| ParkRunning3 $3840 \times 2160$ | 18.5% |
| Average | 12.3% |

### 3.2. Affine Motion Estimation Early Skipping Method

The rate-distortion (RD) cost calculations for interprediction selection are performed recursively in determining the best interprediction for blocks. It considers the strong correlation between the current block and its adjacent CUs in terms of texture and motion information. Furthermore, the best interprediction mode and RD costs for adjacent blocks have been stored in the encoder. Therefore, it is possible to determine whether the current block needs to execute the affine mode based on the encoding information of the previously reconstructed information of adjacent CUs. In addition, the encoding information of the co-located CU in the adjacent frame is also used to determine whether the current block performs affine motion estimation.

Merge mode, affine advanced motion vector prediction (AMVP) mode, and skip mode are included in the interprediction. Among them, skip mode can be regarded as a special case in merge mode. Compared with merge mode, skip mode is a simpler mode. It does not require transmitting the prediction residuals, only the index of the best element in the candidate list. The descriptions of merge mode, AMVP mode, and skip mode are shown in Table 2. Figure 2 shows the CU partition results of the second frame extracted from the sequence "BasketballDrive" under low delay P (LDP) configuration, where the affine mode as the best interprediction is boxed in blue, and the skip mode as the best interprediction is boxed in red. From Figure 2, we can observe that skip mode is mainly used for background areas with a slight texture and slow-moving areas. Affine mode is used chiefly for blocks with dramatic motions and rich textures. Therefore, skip mode, and affine mode are mutually exclusive in most cases. Furthermore, in both affine and skip modes, they are selected with surrounding CUs (e.g., left and up).

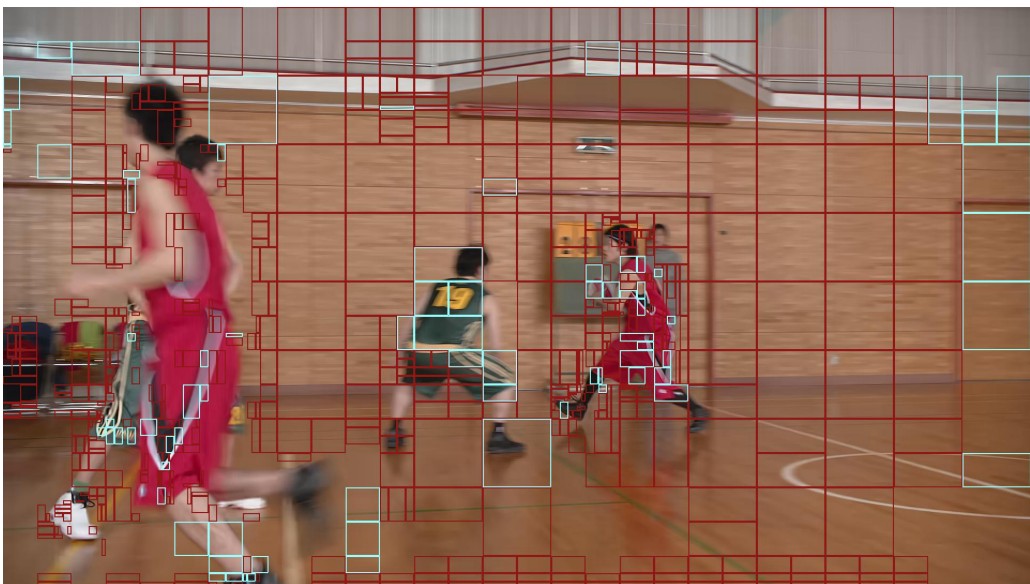

**Figure 2.** The optimal interprediction selected for each CU in "BasketballDrive".

**Table 2.** Details of the three modes in interprediction.

| Mode | Transmission |
|:---:|:---:|
| Skip | index |
| Merge | Index and prediction residuals |
| AMVP | Index, prediction residuals, and MVD |

To further explore the relationship between the encoding information of the reconstructed adjacent blocks and the affine mode used in the current block, the probability of using the affine mode as the optimal interprediction under different conditions in videos of different resolutions is investigated. Specifically, we assume that the event $X_{skip}$ is the adjacent blocks to the left and the top containing the skip mode, whereas the affine mode is not included. Event $Y_{skip}$ represents the co-located CU in the adjacent frame by using skip mode. $P(A|X_{skip})$ represents the probability that the best interprediction for the current block is the affine mode under the event $X_{skip}$ condition as follows,

$$P(A|X_{skip}) = \frac{P(AX_{skip})}{P(X_{skip})}, \tag{4}$$

where $P(AX_{skip})$ is the probability that event $A$ and event $X_{skip}$ will occur simultaneously. $P(X_{skip})$ represents the probability of event $X_{skip}$ occurring. Similarly, $P(A|Y_{skip})$ denotes

the probability that the best interprediction for the current block is the affine mode under the event $Y_{skip}$ condition calculated as

$$P(A|Y_{skip}) = \frac{P(AY_{skip})}{P(Y_{skip})}, \tag{5}$$

where $P(AY_{skip})$ is the probability that event $A$ and event $Y_{skip}$ will occur simultaneously. $P(Y_{skip})$ represents the probability of event $Y_{skip}$ occurring. In experiments, when performing open-source VTM-7.0 to encode video sequences, we record the number of using the affine motion mode as the best interprediction under events $X_{skip}$ and $P(Y_{skip})$ conditions, respectively. The number of events $X_{skip}$ and $P(Y_{skip})$ are also recorded. The conditional probabilities are obtained by their respective corresponding ratios. Table 3 presents statistics on the probabilities of the five video sequences at different resolutions. From Table 3 we can see that the average value of $P(A|X_{skip})$ is not large, at 0.12. This indicates that the probability of the current CU using the affine mode as the best mode is low when the best interprediction for adjacent blocks contains the skip mode and does not include the affine mode. Furthermore, the average value of $P(A|Y_{skip})$ is 0.21, which means there is a small probability that the current block will select affine mode as the interprediction when the skip mode is contained in the co-located CU.

**Table 3.** Statistics on the probability of A event in test video sequences at different resolutions under conditions $X_{skip}$ and $Y_{skip}$.

| Sequences | $P(A|X_{skip})$ | $P(A|Y_{skip})$ |
|---|---|---|
| BlowingBubbles　$416 \times 240$ | 0.14 | 0.20 |
| RaceHorsesC　$832 \times 480$ | 0.10 | 0.17 |
| BasketballDrill　$832 \times 480$ | 0.09 | 0.19 |
| FourPeople　$1280 \times 720$ | 0.13 | 0.24 |
| Cactus　$1960 \times 1280$ | 0.16 | 0.27 |
| Campfire　$3840 \times 2160$ | 0.17 | 0.30 |
| ParkRunning3　$3840 \times 2160$ | 0.23 | 0.36 |
| Average | 0.12 | 0.24 |

Based on the above statistics and analysis, we propose a low-complexity method to skip the affine mode early by utilizing the adjacency encoding information. More specifically, we obtain the best interprediction of three types of CUs, including the co-located CU in the adjacent frame, adjacent left and upper CUs of current block, to determine whether to perform the affine mode after the process of translational motion estimation. Concretely, if the adjacent blocks to the left and above contain skip mode but not affine mode, the affine motion estimation process will be skipped. Furthermore, suppose the co-located CU in the adjacent frame uses skip mode. In that case, the affine motion estimation process for the current CU will also be skipped ahead. The details of the interprediction process, which also included the proposed affine motion estimation early skipping method, are shown in Figure 3.

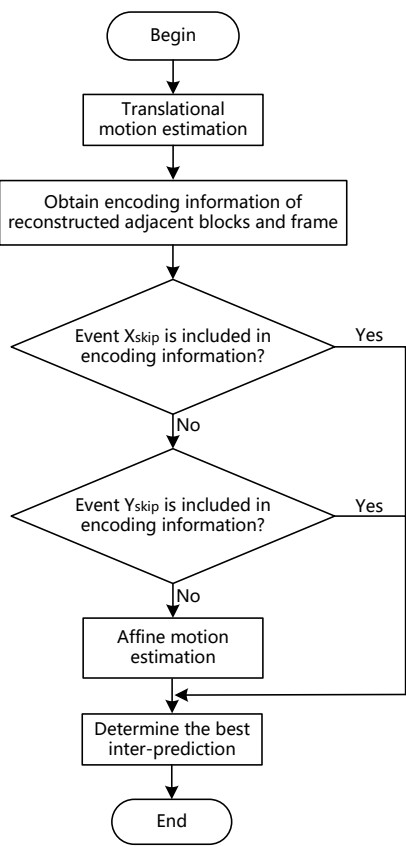

**Figure 3.** The overall framework of interprediction which includes the proposed affine motion estimation early skipping method.

## 4. Experiments and Results Analysis

### 4.1. Experimental Settings

In order to verify the effectiveness of the proposed adjacency encoding information-based fast affine motion estimation method, we implement our method and conduct the experiments based on a VVC test model (VTM-7.0 anchor) under JVET common test conditions (CTC) configurations [38]. The LDP, low delay B (LDB), and random access (RA) configurations are used in experiments, and the quantization parameters (QPs) are set at 22, 27, 32, and 37. We verify the effectiveness of the proposed method through abundant experiments performed on the "computer with Intel core i5-3470 CPU", including the comparisons with the VVC anchor and state-of-the-art methods. The details of experimental environments are shown in Table 4. Furthermore, the details of open-source test video sequences are displayed in Table 5. The Bjøntegard delta bitrate (BDBR) [39] is used to measure the encoding performance of the introduced adjacency encoding information-based fast affine motion estimation method, and the negative value represents performance gains. Moreover, we also use the Bjøntegard delta peak signal-to-noise rate (BD-PSNR) [39] to evaluate encoding performance, where a positive value denotes improved performance. In addition, the overall encoding time savings $SavT_{all}$ of the introduced method compared to the VVC anchor is calculated as follows,

$$SavT_{all} = \frac{T_{ori} - T_{pro}}{T_{ori}} \times 100\%, \tag{6}$$

where $T_{ori}$ and $T_{pro}$ represent the overall encoding time of the original VVC anchor and proposed method. To further demonstrate the effectiveness of the proposed method, we

show the time savings $SavT_{af}$ of affine motion estimation in the proposed method compared to the original VVC affine motion estimation process, which is calculated as follows,

$$SavT_{af} = \frac{T_{afori} - T_{afpro}}{T_{afori}} \times 100\%,$$ (7)

where $T_{afori}$ and $T_{afpro}$ denote the encoding time of affine motion estimation in the original VVC anchor and proposed method.

**Table 4.** The environments and conditions of simulation.

| Items | Descriptions |
|---|---|
| Software | VTM-7.0 |
| Configuration File | encoder lowdelay P vtm.cfg |
| | encoder lowdelay vtm.cfg |
| | encoder randomaccess vtm.cfg |
| Video Sequence Size | $416 \times 240$, $832 \times 480$, |
| | $1280 \times 720$, $1920 \times 1080$, $3840 \times 2160$ |
| Quantization Parameter (QP) | 22, 27, 32 and 37 |
| Sampling of Luminance to Chrominance | $4 : 2 : 0$ |

**Table 5.** Detailed characteristics of the experimental video sequences.

| Class | Sequences | Size | Bit-Depth | Frame Rate |
|---|---|---|---|---|
| A1 | Campfire | $3840 \times 2160$ | 10 | 30 |
| | FoodMarket4 | $3840 \times 2160$ | 10 | 60 |
| A2 | ParkRunning3 | $3840 \times 2160$ | 10 | 50 |
| | CatRobot | $3840 \times 2160$ | 10 | 60 |
| B | BasketballDrive | $1920 \times 1280$ | 8 | 50 |
| | BQTerrace | $1920 \times 1280$ | 8 | 60 |
| | Cactus | $1920 \times 1280$ | 8 | 50 |
| | RitualDance | $1920 \times 1280$ | 10 | 60 |
| C | BasketballDrill | $832 \times 480$ | 8 | 50 |
| | BQMall | $832 \times 480$ | 8 | 60 |
| | PartyScene | $832 \times 480$ | 8 | 50 |
| D | BasketballPass | $416 \times 240$ | 8 | 50 |
| | BlowingBubbles | $416 \times 240$ | 8 | 50 |
| | RaceHorses | $416 \times 240$ | 8 | 30 |
| E | FourPeople | $1280 \times 720$ | 8 | 60 |
| | Johhny | $1280 \times 720$ | 8 | 60 |
| | KristenAndSara | $1280 \times 720$ | 8 | 60 |
| F | Slideshow | $1280 \times 720$ | 8 | 20 |
| | SlideEditing | $1280 \times 720$ | 8 | 30 |
| | BasketballDrillText | $832 \times 480$ | 8 | 50 |

*4.2. Experimental Results and Analyses*

First, the VTM-7.0 with AMC is used as a benchmark comparison to represent the performance of the proposed method. The encoding time savings by the adjacency encoding

information-based fast affine motion estimation method is shown in Table 6. Concretely, the BDBR and BD-PSNR, which measure the encoding performance of the model, are also included. Table 6 illustrates that compared to the standard anchor, the proposed algorithm achieves a low-complexity encoder to decrease encoding time for all tested sequences. We can observe that the proposed algorithm saves averages of 10.11% of the overall encoding time and 40.85% of the encoding time for the affine motion estimation process. The BD-PSNR only reduces 0.006 dB, and BDBR increases by 0.16%, which means that the proposed algorithm barely reduces the encoding performance of the VVC encoder. The affine motion estimation times for the "BQTerrace", "BQMall", "RaceHorses", "Johnny", and "SlideEditing" sequences are all saved by over 50%. This is mainly due to the fact that most of these sequences are background regions and contain mostly translational motion, with only a small proportion of the regions using affine mode. As a result, the proposed method reduces the computational complexity of the affine motion estimation in the background and translational motion regions, thereby significantly reducing the encoding time.

**Table 6.** The proposed method compared to the original VVC experimental results.

| Class | Sequences | BDBR/% | BD-PSNR/db | $SavT_{all}$/% | $SavT_{af}$/% |
|---|---|---|---|---|---|
| A1 | Campfire | 0.20 | −0.010 | 10.32 | 47.62 |
| | FoodMarket4 | 0.16 | −0.007 | 9.17 | 37.36 |
| A2 | ParkRunning3 | 0.22 | −0.011 | 9.48 | 39.28 |
| | CatRobot | 0.17 | −0.006 | 9.83 | 43.72 |
| B | BasketballDrive | 0.18 | −0.007 | 9.04 | 35.27 |
| | BQTerrace | 0.12 | −0.001 | 9.80 | 59.43 |
| | Cactus | 0.17 | −0.004 | 8.37 | 27.45 |
| | RitualDance | 0.15 | −0.003 | 9.13 | 35.82 |
| C | BasketballDrill | 0.07 | −0.005 | 10.75 | 37.22 |
| | BQMall | 0.01 | 0.000 | 10.68 | 50.45 |
| | PartyScene | 0.19 | −0.009 | 9.21 | 40.28 |
| D | BasketballPass | 0.21 | −0.010 | 11.07 | 33.58 |
| | BlowingBubbles | 0.23 | −0.008 | 8.70 | 44.34 |
| | RaceHorses | 0.25 | −0.010 | 9.13 | 50.16 |
| E | FourPeople | 0.21 | −0.008 | 9.23 | 21.76 |
| | Johhny | 0.03 | 0.000 | 15.47 | 54.19 |
| | KristenAndSara | 0.22 | −0.005 | 9.42 | 27.49 |
| F | Slideshow | 0.09 | −0.004 | 11.85 | 49.71 |
| | SlideEditing | 0.06 | −0.003 | 12.27 | 51.86 |
| | BasketballDrillText | 0.24 | −0.011 | 9.32 | 29.74 |
| Average | - | 0.16 | −0.006 | 10.11 | 40.85 |

To further validate the effectiveness of the proposed method, we also compare the adjacency encoding information-based fast affine motion estimation method with the state-of-the-art fast methods. As displayed in Table 7, the proposed method achieves more encoding time savings compared to Ren et al. without a significant increase in BDBR. As we understand it, the reason may be that although the method proposed by Ren et al. reduces computational complexity by optimizing affine motion estimation, it still requires

the execution of affine mode for all CUs. In order to obtain greater encoding time savings, the proposed method is based on the adjacency encoding information to skip most of the affine motion estimation process.

**Table 7.** The proposed method compared to the state-of-the-art experimental results.

| Sequences | Ren et al. [36] | | Proposed | |
|---|---|---|---|---|
| | BDBR/% | $SavT_{all}/\%$ | BDBR/% | $SavT_{all}/\%$ |
| Cactus | 0.11 | 6.00 | 0.17 | 8.37 |
| BQTerrace | 0.04 | 8.00 | 0.12 | 9.80 |
| BasketballDrive | 0.08 | 5.00 | 0.18 | 9.04 |
| BQMall | 0.05 | 5.00 | 0.01 | 10.68 |
| PartyScene | 0.26 | 4.00 | 0.19 | 9.21 |
| BasketballDrill | 0.06 | 3.00 | 0.07 | 10.75 |
| BasketballPass | 0.08 | 2.00 | 0.21 | 11.07 |
| BlowingBubbles | 0.12 | 6.00 | 0.23 | 8.70 |
| RaceHorses | 0.08 | 5.00 | 0.25 | 9.13 |
| Average | 0.10 | 4.89 | 0.16 | 9.64 |

To visualise the impact of the proposed method on VVC compression performance and video quality, the R-D curves for the test sequences "BlowingBubbles" and "BasketballDrill" are presented in Figure 4. The results show that the proposed method almost coincides with the R-D curve of the original VVC, indicating that the adjacency encoding information-based fast affine motion estimation method proposed in this paper does not significantly reduce the compression performance. Furthermore, Figure 5 displays the comparison of the subjective quality from "PartyScene" encoded by the proposed method and the original VVC anchor with QP 27 under RA configuration. We can observe in Figure 5 that the subjective quality difference between the reconstructed frame and the original is virtually invisible to the eyes. In summary, the adjacency encoding information-based fast affine motion estimation method proposed in this paper achieves significant savings in encoding time with negligible subjective quality loss.

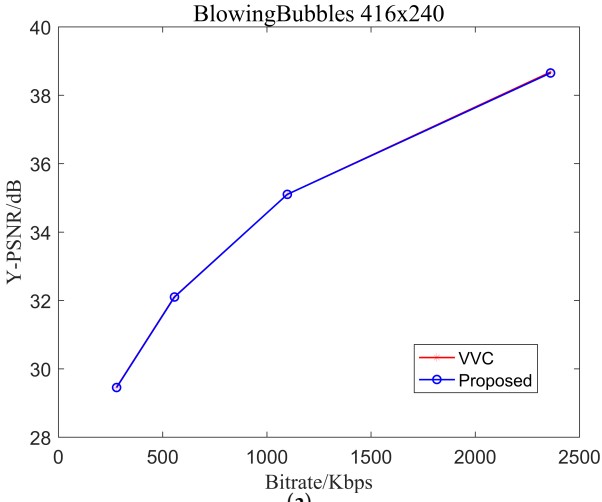
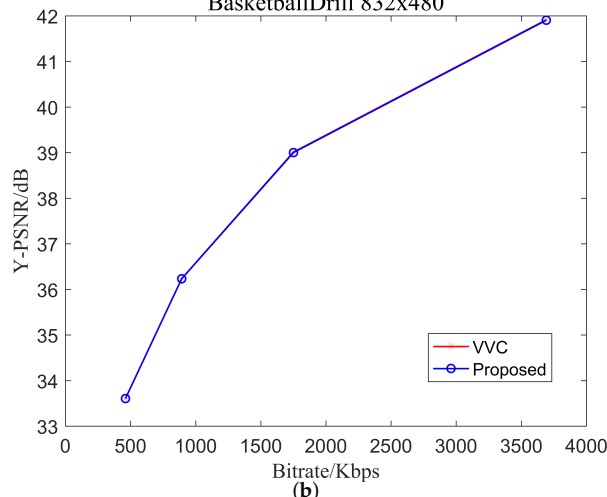

**Figure 4.** The R-D curves of sequences "BlowingBubbles" (Class D) and "BasketballDrill" (Class C) under LDP and RA configurations. (**a**) LDP configuration; (**b**) RA configuration.

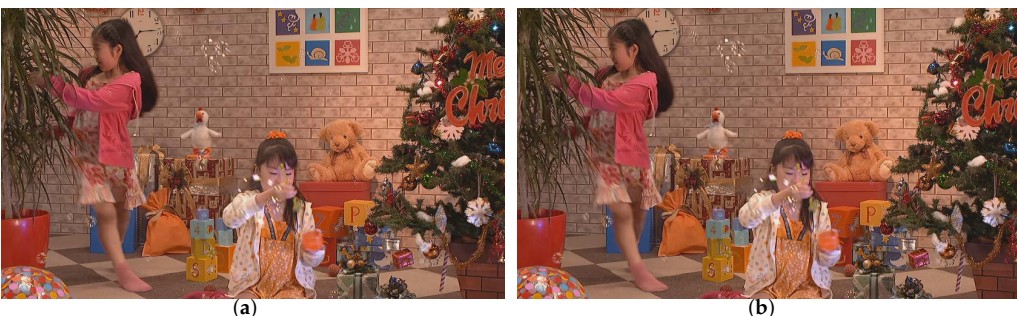

| (a) | (b) |

**Figure 5.** Subjective quality comparison of the 10th decoding frame of "PartyScene" from Class C. (**a**) original VVC; (**b**) proposed method.

## 5. Conclusions

The VVC is unsuitable for use in real-time applications as a series of computationally complex coding tools added. In order to address the above issue, this paper proposes an adjacency encoding information-based fast affine motion estimation method to save time in video coding. We analyze the trade-off between computational complexity and encoding performance improvement by counting the probability of choosing the affine mode as the best interprediction. Moreover, the affine mode calculation process can be skipped in advance by fully using the adjacency encoding information. The experimental results show that the proposed algorithm reduces 10.24% compared to the anchor encoder without significantly degrading the subjective quality of the encoded video. In future work, we will focus on fast CU partitioning methods in VVC and combine them with the proposed adjacency encoding information-based fast affine motion estimation method to achieve more encoding time savings.

**Author Contributions:** Conceptualization, X.L.; methodology, X.L.; software, X.L.; validation, X.L. and J.H.; formal analysis, X.L.; investigation, X.L.; writing—original draft preparation, X.L.; writing—review and editing, X.L., J.H., Q.L., and X.C.; visualization, X.L.; supervision, X.L., J.H., Q.L., and X.C.; project administration, X.L. and J.H.; funding acquisition, X.L.. All authors have read and agreed to the published version of the manuscript.

**Funding:** This research was partially funded by Natural Science Foundation of Hunan, China (2020JJ7063).

**Institutional Review Board Statement:** Not applicable.

**Informed Consent Statement:** Not applicable.

**Data Availability Statement:** Not applicable.

**Conflicts of Interest:** The authors declare no conflict of interest.

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
