# Peer review of "An Adjacency Encoding Information-Based Fast Affine Motion Estimation Method for Versatile Video Coding"

_electronics, doi:10.3390/electronics11213429_

Round 1

Reviewer 1 Report

An early skipping of affine mode based on adjacent frame block and adjacent blocks are proposed.

1. However, more elaboration of the algorithm is required in graphical representation, like flow charts are required. Fig.2 might represent something like this, but the authors missed it in the manuscript.

2. Fig.2 is missing or misplaced, and the authors are suggested to look into manuscript thoroughly before uploading it.

3. Computing platform used for simulation must be highlighted in the results like "computer with intel core i7" or "computer with AMD ..." or any others.

4. Atleast two video sequences of 4k are recommended to use, for validating the algorithm

5. The authors are recommended to include (in table 6. or separately in another table) the results of individual algorithms (like X-skip and Y-skip) along with the combined algorithm results.

6. The authors are recommended to put the values of individual encoding time, PSNR and bitrate values before and after applying algorithm, in a separate table or in table.6.

7. In the original VTM, the authors are suggested to put the affine modes computation time or percentages, for all block sizes in PU for any sequence. Like 8x8 - __%, 8x16__ %, where percentage represents total motion estimation time or encoding time. After applying the algorithm, another column with percentages at PU level, would give a fair evaluation of the overall algorithm.

Round 2

Reviewer 1 Report

The manuscript still misses fig.2 and fig.3, which are essential for the judgement of the paper.

Author Response

Thank you for your helpful comment.

We are sorry that Figure 2 and Figure 3 are missing in the manuscript from your side. In fact, Figure 2 and Figure 3 are included in the manuscript from my side. As we understand, the reason may be that when writing the code related to inserting Figures in the manuscript, the information about where the figures are stored in the file does not correspond to your side. We have modified the relevant part of the latex code. Hope you will be satisfied with our revisions.

Reviewer 2 Report

Authors managed to address most of the reviewers comments. Reviewer still believes that the data set should contain more high resolution sequences and less small resolution sequences, i.e. it should contain sequences class B and higher. However, the results presented are convincing.

Author Response

Thank you for your kind suggestion.

As suggested by the reviewers, we have added the sequences “FoodMarket4” from Class A1, “CatRobot” from Class A2, and “RitualDance” from Class B in the experiments to obtain a more valid statistical result. Table 6 (Section 4.2, Page 9) shows the encoding time savings achieved by the proposed method compared to the original VVC.

Round 3

Reviewer 1 Report

In page, table 3 is shown and it is explained in the above paragraph. The authors should highlight the method or software used to get the individual probability values of Xskip and Yskip. Is it from VTM or any matlab code written seperately, or are they estimated values? it should be highlighted in the explanation.

Author Response

Thank you for your helpful comment.

As suggested by the reviewer, we have added the descriptions in detail to explain how to get the corresponding probability.

For the convenience of reviewers, the amendments are highlighted in red in the revised manuscript.